# The Se–S Bond Formation in the Covalent Inhibition Mechanism of SARS-CoV-2 Main Protease by Ebselen-like Inhibitors: A Computational Study

**DOI:** 10.3390/ijms22189792

**Published:** 2021-09-10

**Authors:** Angela Parise, Isabella Romeo, Nino Russo, Tiziana Marino

**Affiliations:** 1Dipartimento di Chimica e Tecnologie Chimiche, Università della Calabria, Via Pietro Bucci, 87036 Arcavacata di Rende, CS, Italy; angela.parise@unical.it (A.P.); isabella.romeo@unical.it (I.R.); nino.russo@unical.it (N.R.); 2Institut de Chimie Physique UMR8000, Université Paris-Saclay, CNRS, 91405 Orsay, France

**Keywords:** SARS-CoV-2 main protease, DFT, inhibition mechanism, Se–S covalent bond, potential energy surface

## Abstract

The inhibition mechanism of the main protease (M^pro^) of SARS-CoV-2 by ebselen (EBS) and its analog with a hydroxyl group at position 2 of the benzisoselenazol-3(2H)-one ring (EBS-OH) was studied by using a density functional level of theory. Preliminary molecular dynamics simulations on the apo form of M^pro^ were performed taking into account both the hydrogen donor and acceptor natures of the Nδ and Nε of His41, a member of the catalytic dyad. The potential energy surfaces for the formation of the Se–S covalent bond mediated by EBS and EBS-OH on M^pro^ are discussed in detail. The EBS-OH shows a distinctive behavior with respect to EBS in the formation of the noncovalent complex. Due to the presence of canonical H-bonds and noncanonical ones involving less electronegative atoms, such as sulfur and selenium, the influence on the energy barriers and reaction energy of the Minnesota hybrid meta-GGA functionals M06, M06-2X and M08HX, and the more recent range-separated hybrid functional wB97X were also considered. The knowledge of the inhibition mechanism of M^pro^ by the small protease inhibitors EBS or EBS-OH can enlarge the possibilities for designing more potent and selective inhibitor-based drugs to be used in combination with other antiviral therapies.

## 1. Introduction

Since early 2020 the whole world has been trying to face the severe acute respiratory syndrome coronavirus 2 (SARS-CoV-2) [1,2]. With a relatively long incubation period along with symptoms characterized by different levels of severity, the disease is still affecting billions of people and spreading in an unrelenting fashion causing widespread health, social, and economic disruption. Despite originating from the same *Coronaviridae* family as the Middle East Respiratory Syndrome (MERS) and Severe Acute Respiratory Syndrome (SARS) the shared sequence similarity is ~80% [2]. Viruses belonging to the genus Coronavirus are zoonotic and characterized by positive-stranded RNA encapsulated by a membrane envelope of 300–400 nm [3]. The morphology of virions is similar to a crown due to the mushroom-shaped proteins called spike glycoproteins (S proteins) responsible for the host tropism [4]. Human infections caused by the SARS coronavirus are distinguished by the interaction between the S-protein and the human angiotensin-converting enzyme 2 (ACE2); highly expressed on epithelial cells of the respiratory tract [5]. Following host cell entry, the viral single-stranded RNA is released for replication and translation of the virus polyproteins that are processed by two cysteine proteases (CPs), papain-like protease (PL^pro^) and a main protease (M^pro^) also known as 3-Chymotrypsin-like protease (3CL^pro^) [6,7]. M^pro^ acts in the immune regulation and cleavage of the polyprotein at eleven different sites generating nonstructural proteins, such as RNA polymerase, endo- and exo-ribonuclease, which are relevant in the replication process of the virus [8]. Considering that M^pro^ is only found in the virus rather than in the host cell, this protein represents an interesting target for the development of new promising anticoronavirus therapeutic agents [6,7]. Structurally, SARS-CoV-2 M^pro^ forms a homodimer (protomer A and B) and each protomer consists of three domains: domain I (residues 8–101), domain II (residues 102–184), and domain III (residues 201–303) which are connected by a loop region (residues 185–200) [6,7]. In M^pro^, the catalytic dyad consisting of His41 and Cys145 is located in the cleft between domains I and II [9,10]. The main aim of this protein is cleaving the polyprotein pp1a and pp1ab translated from the viral RNA at 16 different positions to generate important structural proteins in addition to nanostructural proteins (NSPs) which are involved in arresting the process of viral assembly in the replication cycle [11]. The presence of a reactive sulphur in the active site of cysteine proteases provides a target for the design of many covalent and noncovalent inhibitors [11,12,13,14,15]. Among the in vitro investigated covalent inhibitors, it was observed that ebselen [(2-phenyl-1,2-benzoisoselenazol-3(2H)-one], (EBS) shows stronger inhibition against the SARS-CoV-2 virus than N3, the known Michael acceptor inhibitor [15,16,17,18,19]. EBS is a low-molecular-weight organoselenium drug having a pleiotropic mode of action and because of its very low toxicity has encountered no barriers for use in humans [19,20,21]. In fact, in the pre-COVID-19 era, EBS was known as a mimic of glutathione peroxidase and inhibits several enzymes involved in inflammatory processes, such as lipoxygenases, NO synthases and NADPH, providing it with remarkable anti-inflammatory, antiatherosclerotic and cytoprotective properties [22,23]. EBS is known to act as potent inhibitor of proteins implicated in the biosynthesis of the mycobacterial cell wall [20,24] by covalently modifying a noncatalytic cysteine through the reversible oxidation of the γ-sulfur of the amino acid by the EBS selenium, while in the case of M^pro^, the cysteine target of ebselen is crucial for its protease activity [25]. In this system there is evidence arising from experimental study that the cysteine residue involved is present in the catalytic site (Cys145) [15]. Furthermore, EBS has also shown preclinical efficacy for cisplatin-induced ototoxicity [26,27] and has been considered for the treatment of bipolar disorders and hearing loss [28]. EBS inhibition properties have been evaluated in different clinical trials, exhibiting its safety in humans due to its extremely low cytotoxicity [29,30] and therefore it provides the promise of drug repurposing. On the basis of these preclinical studies the reaction of ebselen was reported with cysteine residues from completely unrelated proteins including the C-terminal domain of the HIV-1 capsid, Mycobacterium tuberculosis transpeptidase LdtMt2, glutamate dehydrogenase, Clostridium difficile toxins, Mycobacterium tuberculosis antigen 85C enzyme and many others [20,31,32,33,34,35,36]. Ebselen and some of its synthesized analogues were found to inhibit both SARS-CoV-2 M^pro^ and PL^pro^ [37,38]. Repurposing of known drugs can provide an accelerated path for approval and a likely option to address the current COVID-19 pandemic. In particular, EBS has displayed inhibition against the M^pro^ of SARS-CoV-2 virus with the concentration required to produce 50% of the maximum possible effect and indicating antiviral activity in cells, the EC50, equal to 4.67 µM; superior to that of N3 (16.77 µM) [15]. Furthermore, the improvement of the inhibitory potency of the ebselen derivative obtained by the hydroxyl group in the position ortho of the N-phenyl ring (EBS-OH of Figure 1) on the other Achilles’ heel SARS-CoV-“ PL^pro^, induced us to investigate the M^pro^ inhibition mechanism by both EBS and EBS-OH inhibitors [21,39]. The behavior of the heavier chalcogens, S and Se, in biological systems has only recently gained more attention, in particular Se has been shown to form an Se–S intermediate in selenoprotein reductase and formate dehydrogenase. They play a central role in the enzyme’s activation [40,41] and also in deiodinase biomimetics [42]. So far the SARS-CoV-2 M^pro^ inhibition mechanism has been studied by using Michael acceptor and peptidomimetic inhibitors at the QM-MM level of theory [43,44,45,46,47]. So, the mechanistic understanding of the Se–S covalent bond formation promoted by inhibitors, such as EBS and EBS-OH in the present study, could contribute to enlarge the molecular inhibition mechanism of SARS-CoV-2 M^pro^ and stimulate the design of other similar drugs.

Moreover, stimulated by the most recent literature regarding the inhibition mechanism of M^pro^ enzyme [43,45,46,47] we took into account the protonation state of the catalytic histidine (His41) [43,48,49]. Therefore, we have also performed computational simulations by using molecular docking and classical molecular dynamics (cMD) devoted to accurately model the structure and dynamics of M^pro^ in both the Nδ and Nε protonated states of His41 to better evaluate their eventual influence on the covalent inhibition process by EBS. For clarity, M^pro^ in the Nδ form will in the whole manuscript be named M^pro^-HID and M^pro^ in the Nε form will be M^pro^-HIE. To shed light into the inhibition mechanism of EBS and EBS-OH, the potential energy surfaces (PES) for the formation of the covalent complex between SARS-CoV-2 M^pro^ and the two considered inhibitors have been calculated and analysed, considering the effects of the dielectric value, ε = 80 and ε = 4, for the water and protein environments, respectively. In the case of EBS-OH only the inhibition mechanism by Nδ–His41 was performed. The calculated PES for the two inhibitors can be used to determine whether, and to what extent, a covalent inhibitor is reversible or not. This knowledge is important in order to provide in the near future, additional leads for covalent inhibitors obtained by incorporating small-molecules, such as EBS and EBS-OH, into hybrid molecules which also have few side effects. In addition, the ability of other DFT functionals (ωB97X, M06, M06-2X and M08-HX) as single points on the B3LYP-D3 optimized geometries were tested to obtain more accurate data for the energy barriers and reaction energy of the inhibition mechanism driven by the sulfur–selenium covalent bond formation. 

## 2. Results

In an effort to better understand the molecular mechanisms involved in the inhibition of M^pro^, many works have focused on classical Molecular Dynamics simulations with [12,14,25,48,49,50] and without [11,51] inhibitors. In the present investigation cMD simulations performed on apo M^pro^-HID and M^pro^-HIE represent a preliminary step useful to obtain comparative structural and dynamic properties to deeply describe EBS and EBS-OH inhibition mechanisms. 

### 2.1. MD Analysis of the Apo form of M^pro^-HID and M^pro^-HIE

SARS-CoV-2 M^pro^ has been well characterized by crystallography. So far about 1000 entries of X-ray crystal structures for SARS-CoV-2 M^pro^ are present on the Protein Data Bank in an apo form or a complexed one [15,51,52,53,54,55]. To better evaluate the effects on the structural properties in the catalytic pocket, the starting crystal structure with the PDB code 6W63 [56] was used for cMD simulations considering the apo form and the complexed one with EBS in both the Nδ and Nε protonation states of the His41. The Cys145 residue was considered in the neutral form. As previously suggested [49], these investigations are helpful to monitor what happens during the simulation time in the His41-Cys145 catalytic dyad region, allowing us to better rationalize eventual changes in the shape of the inhibitor catalytic site and their possible influence on the explored inhibition mechanism of EBS [48,57].

To compare the dynamic behavior of M^pro^-Nδ and M^pro^-Nε, several properties were taken into account: root mean square deviation (RMSD); RMSD-based clustering; root mean square fluctuation (RMSF), solvent accessible surface area (SASA), radial distribution function (RDF), volume of the binding pocket, salt bridges and the H-bond between residues. Results are reported in Appendix A.

The analysis of the calculated RMSD for the residue pair Cys145-His41 in both the Nδ and Nε forms (shown in Appendix A) allows us to observe that the spatial orientation of the dyad remains constant during the simulation time in the case of His41-Nδ, while a dissimilar trend assumes the RMSD value of His41-Nε. The major fluctuations observed in RMSD for His41-Nε could suggest different conformations of the Cys145 side chain also previously found in the ortholog protease of SARS-CoV [46,58].

A different behavior may also be noticeable in the trend of RMSF (see Appendix A) where it is evident that mainly domain I (residues 1–100) and III (residues 200–300) suffer a major rearrangement revealing higher mobility.

From the superposition of the most representative structure derived from RMSD-based clustering (Appendix A) of the MD trajectory of SARS-CoV-2 M^pro^-HID and HIE (Figure 1), it is possible to see no appreciable difference in the secondary structure except for the chain related to residue His41, that in M^pro^-HID is organized in a longer alpha-helix including Arg40-His41-Val42-Ile43-Cys44 versus Arg40-His41-Val42 present in M^pro^-HIE. In addition, the insets of Figure 1A emphasize the peculiar feature in the catalytic dyad: the altered rotational state of His41 along with the different conformation of the side chain of Cys145. In fact, the C=O–C_α_–C_β_–S_γ_ torsional angle adopts a trans-like conformation (−149.15°) in M^pro^-HID and a gauche-like one (50.69°) in M^pro^-HIE.

Furthermore, the comparison of the electrostatic potential, shown in Figure 1B,B’, calculated for both protonation states of His41 by solving the Poisson–Boltzmann equation as implemented in the APBS code [59], reveals a more marked distribution of negative regions on the surface in proximity to the catalytic site of the SARS-CoV-2 M^pro^ HID. Moreover, the catalytic pocket volume changes dependently on M^pro^-HID or M^pro^-HIE (see Appendix A). Similar active site flexibility has been previously highlighted by Pavlova et al. [48].

### 2.2. Covalent Inhibition

Jin et al. [15], applying a screening strategy on over 10,000 inhibitors of the SARS-CoV-2 virus main protease identified EBS as a good candidate, revealing through tandem MS/MS analysis, its ability to covalently bind to the cysteine 145 residue of M^pro^. Due to the observed stronger inhibition, they did not exclude the possibility that it can also act as a noncovalent inhibitor [15].

As reported in Figure 2 the inhibition usually takes place in two steps. The compound must first bind noncovalently to the target protein, placing its moderately reactive electrophilic selenyl amide moiety, the warhead in ebselen (as circled in Figure 1), close to the nucleophile (Cys145) of M^pro^ and giving rise to the enzyme−inhibitor complex (EI), in which the binding free energy (ΔG_bind_) depends only on noncovalent interactions and is related to the inhibition constant Ki. In the second stage a chemical reaction transforms the EI complex generating the final covalent complex (E-I). In the case of EBS this mechanism is proposed by MS/MS study [15]. Irrespective of the exothermicity, if high reverse reaction barriers are observed the bond formation is effectively irreversible, so k_−2_ will be zero (see Figure 2).

#### 2.2.1. Ebselen (EBS) and Its Derivative (EBS-OH)

The knowledge of the inhibition mechanism at an atomistic level represents a crucial step in the pathway of drug design since it allows us to characterize the reaction path, including characterization of the transition state species which are undetectable at an experimental level. The investigation also extended to EBS-OH which may prove helpful to better understand if this kind of ebselen derivative can improve the selectivity action, a feature actually not present in ebselen [15,16,17,18,19,21]. As above mentioned, the additional hydroxyl group in the position ortho of the phenyl ring resulted in an increase of the inhibitory potency of ebselen by one order of magnitude [21].

The electrostatic potential maps (MEPs) of the two molecules, shown in Figure 2, enable us to visualize the charge distributions and charge related properties of the two molecules.

It is possible to remark that the presence of an OH moiety on the position ortho of the N-phenyl ring of EBS introduces a redistribution of the charge that can be fruitful in the course of the inhibition mechanism. The NBO charge values collected in Appendix A underline a major variation on the selenyl amide moiety. In particular, the nitrogen (N_2_) becomes more negative (−0.661 e vs. −0.602 e in EBS) and the oxygen (O_3_) decreases its charge (−0.597 e vs. −0.618 e in EBS). Furthermore, the charges on C3′, C4′ and C5′ reflect the effect of OH as an activating group on C5′ (ortho) and C3′ (para), while as a deactivating one on C4′ (meta) with respect to EBS.

Inside the catalytic pocket, EBS tends to assume a folded-like conformation with almost isoenergetic results (~0.32 kcal mol^−1^) to the totally planar one (see Figure 2). It is possible to note that the charge distribution suffers a small modification if compared with the extended conformation, but is still different from that of EBS-OH, evidencing again the role played by its hydroxyl group on the N-phenyl ring. 

#### 2.2.2. Determination of the Inhibitor-M^pro^ Complex

The most representative structure of the M^pro^-HID and M^pro^-HIE, obtained from the clustering MDs trajectory (Appendix A), was used in the next docking simulations with EBS and EBS-OH. Details of the procedure used are given in the Electronic Appendix A. 

The best docked poses M^pro^-HID–EBS, M^pro^-HIE–EBS and M^pro^-HID –EBS-OH (shown in Appendix A) were employed, after solvation and MM minimization, as the starting point for building the active site model.

Active site model: The QM cluster model of the active site was derived following a well consolidated procedure [60,61,62,63]. The model includes: Pro39, Arg40, Val42, Ile43, Cys44, Phe140, Leu141, Asn142, Gly143, Ser144, His163, Met165, Glu166, Leu167, the catalytic dyad His41/Cys145, EBS, W1 and W2 (see Figure 3). Following the indications from the literature [46] and from our cMD results (Appendix A), the Asp187 residue was not included in the model since its involvement in a salt bridge with Arg40 precluded its participation in the inhibition mechanism. As usually requested by the QM cluster model [64,65,66], the amino acid residues were truncated as shown in Figure 3, and the coordinates of the related selected atoms were fixed (indicated by stars) during geometry optimizations to prevent unrealistic movements of residues at the active site. In spite of this, the size of the model used in the present study was large enough to grant the flexibility required for the active site groups during the chemical events of the inhibition process. The model thus consists of 312 or 313 atoms including EBS or EBS-OH, respectively. In all the studied systems, the overall charge of the model is zero.

From the plot of solvent accessible surface area at the catalytic pocket (see Appendix A), we can note a smaller area in the case of M^pro^-HID.

From the water radial distribution function (RDF) analysis, obtained as a function of the distance between the water oxygen and side chains of Cys145 and His41 (black and purple line in Appendix A), a different water distribution can be noted in both M^pro^-HID and –HIE, in the first solvation shell of His41, while a similar distribution takes place in that of Cys145. Furthermore, the two water molecules (W1 and W2) are engaged in an extensive network of H-bonds between them and with the His41 imidazole ring. Such behavior confirms that due to the presence of a catalytic dyad (Cys145 and His41) in M^pro^ [8,67,68], and differently from other cysteine and serine proteases where an Asp/Glu acts as the third catalytic residue, a buried water molecule plays a catalytic role.

### 2.3. Inhibition Mechanism

The explored inhibition mechanism of M^pro^-HID (Figure 3) starts with the activation step that takes place from a proton shift between the thiol of Cys145 and the Nε of the imidazole of His41, affording the Cys-His salt bridge present in the intermediate EI’ adduct. 

In this way the highly nucleophilic thiolate anion is formed. This step is very crucial for catalysis performed by CPs [69]. The formed efficient nucleophile -CH_2_S^(-)^ thiolate, is now able to perform the nucleophile attack on the selenium (TS1) with the formation of a new selenylsulfide bond and the concomitant ring opening, also as result of the water-mediated proton delivery on behalf of His41, finalizing the covalent inhibition (E-I). 

Contrary to what occurred in previous theoretical studies on proteolysis and on the inhibition of other CPs [70,71,72] and in agreement with a more recent work [45], both Cys145 and His41 residues of the catalytic dyad are in the neutral form.

Further indications on the protonation states of the M^pro^ residues comes from pKa values predicted using H++ computations [73] (Appendix A). In fact, the pKa of the Cys145 buried in the active site appears to be lowered with respect to its intrinsic pKa ≈ 8.6 at pH = 7 [74], due to interaction with the nearby histidine residue, as has often been found in other CPs for catalytic cysteines [75,76]. Such an alteration of the intrinsic pKa value in the enzyme active site is usually a consequence of the local electronic environment generated by interactions involving the functional groups in the side chains of amino acids as well as the substrate, as documented in many serine protease members [77,78]. Such a scenario is well depicted in the EI optimized complexes referred to for EBS for both the HID and HIE forms (Figure 4), exhibiting the –SH group of Cys145 fruitfully oriented towards the imidazole of His41 and engaged in other different interactions.

In fact, the related noncovalent interaction (NCI) plots (Figure 5) evidence that the region including the catalytic dyad is characterized by the presence of strong and attractive H-bonds, while the van der Waals contributions appeared more pronounced in the case of the HIE form for EBS due to the planar arrangement assumed by EBS, in this case with respect to that in the HID one. In Figure 5 the NCI analysis of EI in the M^pro^-HIE reveals a set of complex interactions between EBS and the amino residues of the catalytic pocket, which arise from a combination of specific hydrogen bonds (atom–atom interactions) with broader surfaces indicative of stabilizing vdW interactions, as a consequence of the planar conformation. In addition, we underline the presence of an interaction between selenium with the thiol group acting as donor of electron density typical of the chalcogen bond. This was more pronounced in the HID form (3.54 Å) than in the HIE one (4.00 Å) as also evidenced by the more extended isosurfaces of the NCI (Figure 5). These findings confirm that noncovalent interactions are of pivotal importance in controlling molecular recognition phenomena. In both M^pro^-HID and HIE the water molecules trapped inside the active site span the distance from the Nε (δ) of His41 and nitrogen of the selenyl amide moiety with typical H-bond values as shown in Figure 4.

When EBS-OH is considered, the torsion angle Φ between two planes twists the biphenyl-like system to be perpendicular (Φ 84.5°), driven also by the formation of an H-bond between the OH substituent with the carbonyl side chain of the Asn142 present in the catalytic pocket (see Figure 5). NCI analysis clearly shows how the twisted conformation of the EBS-OH molecule results in a better fitting of the geometry of the active site. In the catalytic pocket of the M^pro^-HID, the EBS assumes a less accentuated twisted conformation (Φ −42.7°) than that in EBS-OH, but more evident than that in EBS in M^pro^-HIE (Φ −16.8°).

#### 2.3.1. Inhibition Promoted by EBS

Only the inhibition mechanism related to the M^pro^-HID will be deeply discussed, while details referring to M^pro^-HIE are given in the Appendix A. Starting from the EI structure, reported in Figure 4, the first step of the reaction concerns the activation of the neutral cysteine thiol group to generate the thiolate one. This takes place by the formation of the transition state, TS (Figure 6). This transition state clearly depicts the activation of the –SH nucleophile center where the His41 acts as a proton acceptor (H—Nε: 1.68 Å in M^pro^-HID) from the acid counterpart (thiol group of Cys145), S--H: 1.47 Å in HID. Analysis of the obtained imaginary frequency (Figure 7 and Appendix A) confirms this process. The distance between the sulfur and selenium of EBS is reduced (3.20 Å and 3.05 Å) with respect to the reactant EI, where it assumes the value of 4.00 Å and 3.54 Å in M^pro^-HIE and M^pro^-HID, respectively. The related activation energy is found to be 9.1 kcal mol^−1^ in the protein environment (ε = 4) and 7.4 kcal mol^−1^ in water (ε = 80) for the M^pro^-HID (see Figure 6) [79]. The different barriers found in the activation step for M^pro^-HIE (17.9 kcal mol^−1^ in water and 18.5 kcal mol^−1^ in protein, see Appendix A) clearly give a reason for the preferred protonation state (HID) in agreement with previous works [45,46,47].

Therefore, only the inhibition process promoted by the M^pro^-HID has been taken into account in the case of EBS-OH (see Figure 3).

At the end of the activation step, the EI’ structure, is characterized by a CysS^−^/HisH^+^ zwitterionic pair. Irrespective of an HIE or HID form, our calculations for ES therefore support the finding that Cys145 of the M^pro^ of SARS-CoV-2 works in the neutral form and represents the resting state of the enzyme as previously noted in similar systems [71,80].

Looking at Figure 6 and Appendix A (HIE), the EI’ species representing the noncovalent complex and simultaneously the starting point of the inhibition step and having the negatively charged sulphur, is the key player of the covalent bond formation, and now lies at 2.99 (2.78) Å from selenium and at 2.15 (3.41) Å from NδH^+^ (NεH^+^) (see Appendix A). The shorter S–Se distance in the M^pro^-HID accounts well for the greater stability (by about 4 kcal mol^−1^) than the corresponding species in M^pro^-HIE in both considered environments (see Appendix A).

The Se–N “scissile” bond suffers a sensible elongation (2.02 Å for both HIE and HID) making it more prone for its next cleavage. 

Differently from the protease mechanisms by the Michael and peptidomimetic inhibitors investigated in recent works [43,45,46], where the acylation step follows the cysteine activation, in the present case the TS1 describes the covalent bond formation triggered by a series of chemical events occurring simultaneously. These are promoted by proton transfer from the protonated His41 mediated by W1 and W2 water molecules that cause the EBS ring opening described by the elongation of the Se–N bond in favor of the Se–S bond formation, also driven by the already present chalcogen bond (Se–S, 2.78 Å) (Figure 7). The TS1 vibrational mode mainly displays a linear stretching of these proton shifts involving the Nε, OW2 and OW1 atoms. This hydrogen traffic, promoted by water molecules, remarks their catalytic “help”, occupying the place of a third component of the triad in M^pro^ [43,44,46]. Furthermore, the occurring ring opening in TS1 gives rise to a more negatively charged oxygen of the warhead amide moiety as evidenced by the NBO charges trend (Appendix A), although it is not directly implicated in the nucleophilic attack by the sulfur atom of Cys145 as in the proteolytic event. Due to the more negative charge assumed by the EBS oxygen (Appendix A), the carbonyl moiety engages in an H-bond with the NH of the side chain of Asn142 (Appendix A) remembering the “oxyanion hole” effect in proteases. The barrier to be overcome by EI’ for affording the covalent complex E-I, is 9.8 kcal mol^−1^ and 8.6 kcal mol^−1^ for ε=80 and ε = 4, respectively (Figure 6) and compares well with the analogous S–Se bond formation previously calculated [42,81,82,83].

The obtained barriers however do not change the fact that the kinetics of the inhibition by EBS in M^pro^-HIE is governed by the high barrier of the activation step (see Appendix A), contrary to what is observed for the Michael and peptidomimetic-like inhibitors [43,45].

The final covalent product (E-I) accounts for the arrangements observed in TS1, where the S–Se bond is formed (2.43 Å), the Se–N is now completely broken (3.38 Å) and the neutral His41 is restored (Figure 7). The “selenosulfide intermediate nature” of the covalent product in this case, represents the stable end point, although the S–Se bond length is a little bit longer than that present in the literature for biomimetics [40,84,85]. This finding could be a consequence of a series of features created by the protein microenvironment (i.e., the weak interactions present in the catalytic pocket).

#### 2.3.2. Inhibition Promoted by EBS-OH

On the basis of the obtained PES values reported in Figure 6, the activation step for EBS-OH requires an energetic amount of 8.9 kcal mol^−1^ and 9.3 kcal mol^−1^ in ε = 80 and ε = 4, respectively that is comparable with those obtained in the case of EBS (Figure 6). As previously evidenced in the starting structure EI, the TS also exhibits a distinctive interaction of EBS-OH with the carbonyl of Asn142 (1.67 Å) using the OH moiety as a “hook” for anchoring to the side chain of the amino acid with the twisted conformation assumed by EBS-OH (Appendix A).

As a result of the activation step, the EI’ noncovalent complex is formed showing the sulphur anion of the CysS^−^/HisH^+^ zwitterionic pair at 2.81 Å from its parent atom (Se) accompanied by a consequent elongation of the selenium–sulfide bond (2.05 Å) (Figure 8). The tight interaction with Asn142 mentioned above represents an important factor for the stabilization of this complex with respect to the corresponding one for EBS. In fact, for EBS-OH, the results for this species results are more stable by 3.3 kcal mol^−1^ (4.4 kcal mol^−1^) in a water (protein) environment; related to the starting complex with the neutral Cys145 and His41 (Figure 6) and, as already revealed the formation of the noncovalent complex is an exothermic process. This result represents a distinguishing aspect of EBS-OH with respect to EBS, suggesting the zwitterionic form of the dyad as the most preferred one in the catalytic site of M^pro^-HID. The exothermicity of the EI’ formation evidences a better affinity of EBS-OH towards the catalytic pocket of M^pro^, strictly related to the inhibition constant as above mentioned for Figure 2, and this can also help to explain the greater efficiency of EBS-OH than EBS [21]. Observing the related EI’ optimized structures for EBS (Figure 7) and EBS-OH (Figure 8) the only difference can be ascribed to the presence of the OH moiety that is involved in an H-bond with the carbonyl of Asn142 during the whole examined mechanism (Appendix A). This aspect reduces the mobility of EBS-OH versus EBS in the catalytic pocket. So, the greater stability of the EI’ for EBS-OH (see Figure 6) can justify the improved inhibitory potency observed [21].

Similarly to what occurred in the M^pro^-HID inhibition promoted by EBS, the process continues with the nucleophilic attack of the Sγ atoms on the selenium one and the proton transfer from the catalytic histidine to the nitrogen atom is also mediated by water molecules as depicted in the TS1 (see Figure 8). The optimized geometry offers a structure close to the covalent complex (E-I) owing to the formation of a S–Se covalent bond (2.57 Å) in comparison with the analogous complex in EBS (2.77 Å) (Figure 7 and Figure 8). In agreement with other studies devoted to the inhibition process of M^pro^ by species different from our organoselenium compounds, therein investigated [43,45,47], TS1 is the rate limiting step since the barrier results, calculated to be 18.1 kcal mol^−1^ (16.4 kcal mol^−1^) using ε = 80 (ε = 4), are related to the previous noncovalent complex EI’ (see Figure 6).

In the E-I (Figure 8) the S–Se covalent bond is 2.43 Å while the distance of selenium from the nitrogen is now 3.38 Å. The W2 and W1 molecules are reorganized to establish the H-bond interactions with the inhibition product. From the NCI plots (Figure 9) it is also possible to evidence more extended isosurfaces attributable to vdW contributions as a consequence of the presence of the polarizing OH group with respect to EBS.

For both the considered inhibitors at the end of the inhibition process, the His41 presents a different tautomeric form from the starting complex (Figure 2 and Appendix A, Figure 6, Figure 7 and Figure 8 and Appendix A). The strong stability of the E-I final complex for the covalent S–Se bond formation accounts well for the lack of observed reactivation of enzymes treated by ebselen [21] so, though the conversion between His tautomeric states is not a slow process, in the present studied inhibition it is not enough to recover the enzyme.

The inhibition rate constant of SARS-CoV-2 protease by these compounds has not been experimentally determined so it is not possible to compare the calculated barrier with any observed data. From the inhibitory potency investigated for EBS-OH with PL^pro^ SARS-CoV-2, an IC50 constant value of 236 nM was found versus 2 μM for ebselen. We suppose that a similar behavior could take place in the case of M^pro^ of SARS-CoV-2 with a half-maximal inhibitory concentration of EBS-OH better than that of EBS (0.67 μM) [15]. On the basis of our results, we would expect a smaller value of EC50 for EBS-OH and therefore a more potent drug.

As can be seen from PESs (Figure 6), and the energy profiles for both EBS and EBS-OH do not suffer drastic variations based upon the effects of the two considered environments (protein and water). This is not surprising if a larger portion of the enzyme treated quantum mechanically (314 atoms) is considered. Indeed, careful systematic studies have widely demonstrated that the solvation effects of the surrounding enzyme environment decrease rapidly and even almost vanish in models with a size of around 200 atoms [61,62,64]. On the basis of our experience [61,62,82,83], and because of the comparative nature of the present study, the QM/MM methodology is not expected to introduce a big perturbation on the mechanism but mainly an improved energetic behavior.

That said, some deviation occurred in the case of ε = 4 that proposes for both inhibitors a minor exothermicity of the covalent product (see Figure 6). Probably the dielectric constant value of the water tends to evidence the effects of the –OH moiety.

Since the reversibility of the reaction depends on the barrier of the reverse reaction, and with the inhibition process promoted by EBS and EBS-OH showing comparable activation barriers, the process with more exothermic results will also be more irreversible. On the basis of the described behavior of EBS and EBS-OH in M^pro^-HID so far, our analysis leads us to propose [21] EBS-OH to be an irreversible inhibitor stronger than EBS. The assumption that the reactions of irreversible inhibitors are characterized by exothermicity higher than −22 kcal mol^−1^ [86] emphasizes our findings.

Starting from the principle that no universal rule exists for the choice of the “best” functionals, the B3LYP results for the inhibition process were benchmarked against the latest-generation density functionals, such as M06-2X, ωB97X, M06, M06-2X and M08HX in order to point out an adequate level of theory to use in future investigations on the inhibition promoted by ebselen derivatives. Results are collected in Table 1. Single-point calculations were carried out on the B3LYP optimized geometries for the M^pro^-EBS system and the whole benchmark was limited to the ebselen inhibitor. In fact, in the case of EBS-OH, the single point calculations with the above mentioned functionals were performed only on the EI and EI’ complexes due to the marked behavior of the non-covalent complex (EI’) with respect to that of EBS, with its implications for the future design of possible inhibitors with improved kinetics.

In general, it is noteworthy that the first step of the reaction leading to the noncovalent complex is the most sensitive to the used functionals, in particular for M08HX and ωB97X, in terms of barrier heights which become low enough to suggest that the activation for obtaining the zwitterionic catalytic dyad takes place more easily and regardless of the dielectric constant value (see Table 1). All the functionals, indeed, propose a very similar stability of the EI’ to that of the EI starting complex, suggesting that both the forms could be responsible for the next inhibition step.

The same does not apply for the second step of the reaction, since from the results reported in Table 1 it emerges that the B3LYP functional provides a good description of the energetics of this kind of process. This finding also gives confirmation that the effect of the dielectric constant value on the DFT cluster model calculations saturates well with increasing the system size [87], thus the cluster used appears to be adequate for this system, providing information that should be exploited to generate good candidate SARS-CoV-2 inhibitors.

The ωB97X functional (Table 1) results show it to be the best performing one as known in the literature [63,84], but this does not significantly affect our findings.

In the case of EBS-OH, the exothermicity of the formation of the EI’ complex is confirmed by the used functionals (see Table 1) corroborating the outcomes arising from the B3LYP investigation.

## 3. Computational Methods

As far as the molecular dynamics simulations are concerned, a detailed description is given in the Appendix A. Details on docking protocol procedure including figures (Appendix A) and table (Appendix A) are given in the Appendix A. The best docked pose shown in Appendix A has been used as the starting structure for the inhibition mechanism at the QM level, and geometry optimizations were performed with a B3LYP/D3 functional [88,89,90,91] and 6-31+G basis set for all the atoms as implemented in the Gaussian 09 package [92]. To quantify the ZPE corrections, frequencies were calculated at the same level of theory, excluding the contributions of frozen atoms in the vibrational analysis [93]. To evaluate the environment effects, single-point calculations B3LYP-D3/6-31+G(2d,2p) in the framework of the SMD [94] model were performed on the optimized geometries by using the dielectric constants for protein (ε = 4) and water (ε = 80). The final energies reported are solvation energies arising from single point calculations with the larger basis set corrected for ZPE. In order to test if the 6-31+G basis set used was adequate for describing the inhibition phase, we conducted further calculations using 6-311+G(2d,p) and AUG-cc-pVTZ basis sets and the results are reported in Appendix A. These basis sets confirm the results with the smallest one, since the obtained energy barriers of the inhibition process remain essentially unchanged.

Other latest-generation functionals explicitly developed for systems with long-range electron correlations were employed on the M^pro^-HID inhibition process promoted by EBS for benchmarking, such as M06 [95], M06-2X [95], ωB97X [96], and M08HX [97], which have been recently shown to be among the best-performing density functionals in the calculations on many systems [84,85,98,99,100]. The M06, M06-2x and M08HX functionals usually offer good results for main-group chemistry, including thermochemistry, excitation energies, barrier heights, and noncovalent interaction energies with a better performance of M08-HX for the latter ones [101]. ωB97X belongs to the long-range-corrected functionals [96] and was revealed to be useful to better discriminate the chemistry at stake in the two phases of the inhibition mechanism. It is known that the cluster model presents some limitations, such as that a single-conformation QM cluster approach does not explicitly take into consideration the potential influence of different enzyme conformations on the reaction energetics. There is therefore, a consequent reliance on the starting structure and the use of an implicit solvent model to simulate the cluster’s environment, as discussed very deeply in previous excellent papers [7,10,64,102]. However, it was revealed to be appropriate to study enzymatic reactions, also including their inhibition processes [60,61,62,65,81]. NBO [103] analysis was carried out on all the stationary points intercepted on the PESs and the related results are collected in Appendix A. All the results related to the QM investigation on M^pro^-HIE are reported in the Appendix A.

## 4. Conclusions

In the present work, the covalent inhibition of the main protease from SARS CoV-2 by repurposing organoselenium compounds, such as ebselen and its derivative on the position 2 of benzisoselenazol-3(2H)-one ring, was investigated at the QM level in the framework of density functional theory. By using a large cluster of 314 atoms our quantum chemical computations have clarified, at an atomistic level, the inhibition mechanism of two ebselen-like non-Michael acceptor covalent inhibitors of M^pro^. This enzyme plays a central role in the SARS-CoV-2 viral life cycle and for this reason became an attractive drug target against the COVID-19 disease. The inhibition process occurs in two steps: Cys145 is activated by His41, forming the ion pair Cys-His. The formed nucleophile species (deprotonated Cys145) performs the nucleophile attack on the selenium with the formation of the selenylsulfide bond, and takes place in concert with the ring opening promoted by the water-mediated proton delivery by His41, leading to the stable covalent enzyme-inhibitor complex. The S–Se covalent bond formation, the focus of covalent inhibition and a common step of the two examined inhibitors, implies a series of the chemical events that are caught well in the stationary points intercepted through potential energy surfaces. For both inhibitors, the process result is irreversible even if it is more accentuated in the case of EBS-OH. Our outcomes allow us to suggest that, as already found in PL^pro^ SARS-CoV-2, in M^pro^ SARS-CoV-2, EBS-OH may result in being a more potent drug. Our calculations have provided deeper insights on the observed inhibitory potency of EBS-OH with respect to EBS as presented in the greater stability of the EI’ complex relative to that of EBS. The cluster used appears to be adequate for the examined inhibition, providing information that should be exploited to generate good candidate SARS-CoV-2 inhibitors. Our results could contribute to increase the current knowledge of small-molecule covalent inhibitors and stimulate the design of drugs obtained by their incorporation into new hybrid non-Michael acceptor inhibitors. Although the winning strategy such as vaccines is available against the virus, the resource of antiviral drugs represents a helpful approach to reduce the signs of the disease caused by SARS-CoV-2 or by its variants.

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
