# Peer review of "The Se–S Bond Formation in the Covalent Inhibition Mechanism of SARS-CoV-2 Main Protease by Ebselen-like Inhibitors: A Computational Study"

_ijms, 2021, doi:10.3390/ijms22189792_

Round 1
Reviewer 1 Report
In the manuscript titled “The Se-S bond formation in the covalent inhibition mechanism of SARS-CoV-2 main protease by ebselen-like inhibitors: A computational study” authors used density functional level of theory to study inhibition of main protease of SARS-CoV-2 by ebselen and its analog. Overall, the work is well designed and executed. Few comments are listed below:
- Did you try with different basis sets? It would be interesting if you can add 1 or 2 basis sets in your study
- Molecular docking results need to be explained in detail. In order to validate your docking protocol. You can do re-docking and check the RMSD difference between crystal and dock conformation
Author Response
Reply attached

Reviewer 2 Report
By using density-functional theory, this paper studies the inhibition mechanism of SARS-CoV-2 Mpro by small molecule EBS and its analog EBS-OH. In order to accurately elucidate the interaction between EBS/EBS-OH and Mpro, the authors conducted computational simulation to generate two docking model based on the protonated states of H41. Subsequently, the detailed inhibition process was elucidated and EBS-OH was suggested to be more potent than EBS. So far, most of the small molecule inhibitors targeting Mpro are covalent, this approach would be useful to be applied to optimize the potency of other inhibitors. After throughout evaluation of the research article, I personally felt that the presented article is good due to the research work address an interesting topic for researchers. Some comments and suggestions are presented below to improve quality and to clarify some information.
1. I believe there are over 200 entries of X-ray crystal structures of SARS-CoV-2 Mpro presented in the PDB, can the authors double check with this data?
2. Please provide the reason why the authors chose 6W63 as a computational model.
3. I observed a side-chain conformation change of H41 between Mpro-HID and Mpro-HIE. Does any crystal structure has similar conformation of H41 to the simulated results?
Author Response
Thank you for your comments.
I believe there are over 200 entries of X-ray crystal structures of SARS-CoV-2 Mpro presented in the PDB, can the authors double check with this data?
We checked the available data until now, and we corrected in the text from 200 to 1000 structures.
Please provide the reason why the authors chose 6W63 as a computational model.
SARS-CoV-2 main protease protein structure, along with small-molecule inhibitor (PDB ID: 6W63) (Mesecar et al., 2020). At the beginning of our work, 83 structures were available in PDB related to SARS-CoV-2 main protease. The majority of these structures were bound with a small fragment and were suitable for the fragment-based drug discovery approach. Based on available information on bound inhibitors, missing residues, and resolution of the structure, we decided to select the 6W63 structure for our studies, also due to its R-value observed equal to 0.157, thus ensuring the goodness of the model.
I observed a side-chain conformation change of H41 between Mpro-HID and Mpro-HIE. Does any crystal structure has similar conformation of H41 to the simulated results?
We thank the referee for the observation about side-chain conformation change of His41 between Mpro-HID and Mpro-HIE that after protonation obtained by applying H++ software and MD simulations show between them a slightly different orientation of the imidazole ring, as evident in the Figure 1 of the main manuscript. With the aim to clarify what required by the referee a superposition of different Mpro crystal structures (PDB codes: 6W63, 7JUN, 6LU7, 6Y2G and 6LZE), as reported for clarity in the attached figure (Figure-superposition.pdf), showing as the His41 side chain results to be well comparable. The choice of these PDB all deposited in 2020 is mainly related to their employment in the present work and in other multiscale theoretical studies as below specified:
- PDB 6W63 used in our study
- PDB 6LU7 ref. (44)
- PDB 6Y2G ref. (43)
- PDB 6LZE ref. (46)
- The PDB 7JUN is obtained by X-Ray and neutron diffraction and shows the hydrogen on both Nd and Ne
The Mpro-HIE shows the His41 imidazole ring that deviates more from the other ones and this can influence the different behaviour obtained in the inhibition mechanism. However, conformations resulting from our Molecular Dynamics simulations for Mpro-HID and Mpro-HIE derive from relaxed protein structure in the explicit solvent water at 310 K and therefore describe a different situation than that imposed by the rigid crystallographic packing. Other multiscale theoretical studies, which correspond to references 43, 46 and 48 of the main manuscript, show and highlight this different orientation of the His41, in agreement with our work.
